# Factor Structure of Student Science-Learning Motivation: Evidence from TIMSS U.S. Data

Fa Zhang [1,2,*], Christine L. Bae [2], Michael D. Broda [2] and Alison C. Koenka [3]

1   Department of Education, Open University of China, Beijing 100039, China
2   Department of Foundations of Education, Virginia Commonwealth University, Richmond, VA 23284, USA; clbae@vcu.edu (C.L.B.); mdbroda@vcu.edu (M.D.B.)
3   Jeannine Rainbolt College of Education, The University of Oklahoma, Norman, OK 73019, USA; koenkaac@ou.edu
*   Correspondence: zhangf22@alumni.vcu.edu

**Abstract:** We investigated the structure of science motivation among a nationally representative sample of grade 8 students in the United States from the TIMSS dataset. Drawing on an integrative conceptual model of motivation, the following constructs from situative expectancy-value theory (SEVT) and self-determination theory (SDT) were examined through confirmatory factor analyses, including self-concept, task value (intrinsic and utility value), and sense of relatedness, to test the underlying factor structure of motivation. Internal validity evidence was established, which showed that a four-factor model fits the data significantly better than a three-factor model and one-factor model. This study contributes to the current literature by providing evidence of the four dimensions of motivation in a manner that is theoretically aligned with SEVT and SDT, and validated using the TIMSS dataset. The student's sense of relatedness as a key interpersonal facet of learning motivation is highlighted in this study.

**Keywords:** situative expectancy-value theory; integrative approach; model testing; self-determination theory; science-learning motivation

## 1. Introduction

Student motivation, or the drive that sustains students' attention and engagement in learning [1] (Schunk and DiBenedetto, 2020), continues to be a major focus of national and international research and policy. A large body of literature demonstrates motivation as a crucial antecedent to students' academic achievement in science [2–4] (Pintrich, 2003; Wigfield and Eccles, 2002; and Zhang et al., 2021). Further, findings from studies using national or international datasets to examine the link between student motivation and desired educational outcomes provide notable value, given that large, representative sample sizes facilitate more robust inferences about patterns at the national or global scale e.g., [5–7] (e.g., Di Chiacchio et al., 2016; Grabau and Ma, 2017; and Liou, 2017). Specifically, a substantial body of research, using the Trends in International Mathematics and Science Study (TIMSS) dataset, has accumulated in the last decade that has examined students' motivation in science e.g., [8–12] (e.g., Berger et al., 2020; Guo et al., 2017, 2018; and Wang and Liou, 2017, 2018). Although these findings point to statistically significant positive relationships between student motivation factors and their science achievement, findings from a recent systematic literature review of TIMSS motivation studies show that items chosen to measure the motivation factors vary considerably across studies. Specifically, studies vary in terms of the definitional alignment between the items and the corresponding motivation construct, as well as between the constructs and theoretical framework(s) presented [13] (Zhang and Bae, 2020).

These findings point to the need for a close examination, both empirically (e.g., internal consistency and factor structure) and theoretically (e.g., alignment to established definitions

and frameworks), of motivation constructs developed using items in the TIMSS dataset. Establishing the factor structure and theoretical soundness of motivation constructs is important in order to allow researchers to compare findings across TIMSS studies. Further, the lack of validation of the factor structure for motivation constructs used to estimate inferential models (e.g., multiple regression model and structural equation models) in the current TIMSS literature is problematic because, as a prerequisite, gathering measurement validity evidence on the internal structure is essential to performing inferential analyses [14] (Kline, 2015).

Inspired by the need to contribute a more nuanced understanding of the empirical and theoretical nature of motivation constructs in the context of TIMSS, we examined the motivation factor structure of students' ratings obtained from the TIMSS student questionnaire by taking an integrative motivation perspective [15] (Linnenbrink-Garcia and Wormington, 2019). In doing so, we drew upon two motivation theories prominently cited in existing TIMSS studies [13] (Zhang and Bae, 2020). Thus, the main purpose of this study was to test the factor structure of motivation using theoretically aligned items from the TIMSS dataset. In the following sections, we presented the gaps among motivational constructs in TIMSS, conceptualized motivation in an integrative approach, and reviewed the two leading motivational frameworks used in the study.

### 1.1. Discrepancies in How Motivation Constructs and Theories Are Presented in TIMSS Studies

Various motivation constructs and frameworks have been cited across TIMSS studies; however, the items used to specify the constructs and the theoretical alignment among constructs and items vary widely across studies (see [13] Zhang and Bae, 2020 for a review). For instance, drawing from SDT [16] (Ryan and Deci, 2000), Lay and Chandrasegaran (2016) [17] conceptualized students' motivation as confidence in science (i.e., self-efficacy or self-concept), with nine items from the TIMSS questionnaire, intrinsic motivation consisting of five items, and extrinsic motivation consisting of six items. In contrast, another TIMSS science motivation study that also drew on SDT conceptualized motivation as science self-efficacy specified by a different set of five items, and intrinsic motivation consisting of 13 items (Leong et al., 2018) [18]. In addition to different constructs being used under the same framework across studies with varying degrees of theoretical alignment, the number and nature of items used to measure the same motivation constructs also varied across studies. Taking self-belief-related factors as an example, science self-concept was constructed with five (e.g., Guo et al., 2018) [10], four (e.g., Liou, 2014; and Tsai and Yang, 2015) [19,20], or nine (e.g., Wang and Liou, 2017, 2018) [11,12] different motivational items across studies. As another example, confidence in science consisted of four (e.g., Gao, 2014) [21], six (e.g., House and Telese, 2017; and Liu and Wang, 2019) [22,23], or nine (e.g., Akilli, 2015; and Lay and Chandrasegaran, 2016) [17,24] different items. Similar inconsistencies also existed for other motivation constructs, such as intrinsic value, interest, intrinsic/extrinsic motivation, attitude towards science, utility value, and general value in science (e.g., Akilli, 2015; Liou, 2017; and Liou and Liu, 2015) [7,24,25]. Taken together, because of the theoretical misalignment among motivational constructs and items across TIMSS studies, there is a need to examine an integrative measurement model.

Based on a systematic literature review of motivation items, constructs, and related frameworks in TIMSS studies [13] (Zhang and Bae, 2020), the current study aims to empirically test an integrative measurement model that draws together SDT and SEVT constructs that are best represented by items on the TIMSS student questionnaire. This study contributed to the small line of research on examining middle school students' science-learning motivation in the context of an international large-scale assessment. Additionally, results from a recent study examining measurement models underlying motivational constructs with TIMSS data from 58 countries showed that, although self-beliefs and intrinsic motivation scales are assumed to be unidimensional, the single-dimension CFA model failed to satisfactorily fit in all 58 country cases [26] (Pedrero and Manzi, 2020). Their results also showed cultural biases in response patterns, showing that only the motivation scale

in Science and Mathematics presents an acceptable level of invariance to produce valid comparisons across the different countries. We extend their findings by examining the utility value (i.e., the usefulness of learning) and sense of relatedness (i.e., belonging to schooling), in addition to self-beliefs and intrinsic motivation (conceptualized as intrinsic value in this study) in science learning. In doing so, we provide a more comprehensive model of motivation.

### 1.2. Integrative Approach to Conceptualizing Motivation

It is important to integrate motivation frameworks that comprehensively account for the different types of motivation in science [15] (Linnenbrink-Garcia and Wormington, 2019). Representing the complex student drives in science learning, motivation has recently been conceptualized as multifaceted [27,28] (Chen et al., 2019; and Dweck, 2017). That is, concentrating on a single motivation theory often fails to take into account how different motivation factors influence students' learning interactively. As a result, doing so contributes to the mismatch between competing motivation research that is siloed by a particular theory or tradition, and the experiences that teachers report regarding the complex nature of student motivation in their classrooms [15] (Linnenbrink-Garcia and Wormington, 2019).

In line with these contemporary efforts [15,27,28] (Chen et al., 2019; Dweck, 2017; and Linnenbrink-Garcia and Wormington, 2019), to provide a more complete and nuanced picture of how students are motivated in science learning, we integrate motivation constructs across SEVT and SDT in the current study. These frameworks were selected because SEVT has been shown as the most frequently used framework in TIMSS science motivation studies [13] (Zhang and Bae, 2020). SDT is integrated to account for student interpersonal motivation in learning (i.e., sense of relatedness) and has also been cited in past TIMSS studies (e.g., Leong et al., 2018) [18]. These motivation theories and corresponding constructs are reviewed next.

### 1.3. A Brief Review of Motivational Frameworks and Self-Beliefs

#### 1.3.1. Situated Expectancy-Value Theory (SEVT)

As a developmental model of learning motivation, SEVT proposes that students' competence beliefs, expectancy for success, and task values predict students' academic learning motivation and pursuit of achievement goals (Eccles and Wigfield, 2020; and Wigfield and Eccles, 2002) [3,29]. Notably, we drew from SEVT throughout the study instead of the Expectancy-Value Theory (Wigfield and Eccles, 2002) [3] to acknowledge the situated nature of the theoretical tenets.

There are different self-belief- or competence-belief-related constructs, such as self-concept and self-efficacy (detailed further below). With regard to the task values in SEVT, the intrinsic value represents enjoyment when performing an assignment, and the utility value, which is also called usefulness, refers to how a task fits into an individual's plans (Eccles and Wigfield, 2020; and Wigfield and Eccles, 2002) [3,29]. Several TIMSS items in the student science questionnaire directly tap into students' task values (e.g., intrinsic value: "I learn many interesting things in science" and utility value: "I need to do well in science to get the job I want"). In this study, we examined whether students value learning science as a general task value construct (i.e., task value) or distinguish between enjoying learning science (i.e., intrinsic value) and thinking learning science is useful and important (i.e., utility value).

#### 1.3.2. Self-Determination Theory (SDT)

Another leading motivational framework, SDT, proposes that competence, relatedness, and autonomy are three basic psychological needs that underlie intrinsic motivation, or doing something for one's inherent interest and enjoyment (Ryan and Deci, 2000, 2020) [16,30]. As a broader framework of human motivation, SDT articulates intrinsic motivation and varied extrinsic sources of motivation that play important roles in students' learning behav-

iors. Autonomy refers to the need to feel like one has choice and agency in one's learning process (Ryan and Deci, 2000) [16]. Similar to other self-belief constructs, competence refers to a feeling or a sense of confidence and ability to accomplish a task well. Finally, relatedness, often referred to as a sense of belongingness, is the need to feel connected to others (Ryan and Deci, 2000) [16]. Based on the items available in the TIMSS student questionnaire, we focus on the relatedness construct in this study. Past studies have shown that a sense of relatedness, such as perceiving teachers as being caring or having positive relationships with peers, facilitated students' motivation to engage in learning science and science achievement (Marshik et al., 2017; Martin et al., 2016a; Ryan and Grolnick, 1986; and Zhang et al., 2021) [4,31–33]. Thus, drawing from SDT, relatedness, a fundamental interpersonal need that supports students' motivation to learn in school, was included and examined in this study as a key motivation construct.

### 1.3.3. Self-Concept and Self-Efficacy

There are many constructs related to students' ability, concepts or beliefs that have been used in TIMSS science motivation studies, including confidence, self-concept, and self-efficacy (Zhang and Bae, 2020) [13]. In the present study, self-concept was used to represent students' beliefs in their science-learning ability. We chose self-concept because it refers to the evaluation of one's own general ability in a domain (Marsh and Martin, 2011) [34], compared to self-efficacy, which represents one's perception of one's ability to successfully complete a specific academic assignment or achieve an academic goal (Bong and Skaalvik, 2003; and Pajares, 1996) [35,36]. That is, self-efficacy focuses on specific perceptions of whether an assignment can be completed successfully or a goal can be reached (Jansen et al., 2015) [37]. However, belief-related items in TIMSS student science questionnaires (e.g., "I usually do well in science.") focus more on the general science subject domain rather than a specific assignment, such as completing an experiment. As a more global construct referring to general ability, student self-concept in science was included and examined in the measurement models.

### 1.4. Baseline vs. Three-Factor vs. Four-Factor Motivation Structure

In this study, we systematically tested a one-, three-, and four-factor structure of motivation. The one-factor model was used as a baseline model (e.g., Ben-Eliyahu et al., 2018; and Brown, 2015) [38,39]. We hypothesized that motivation is multidimensional. Specifically, taking an integrative approach that draws on SEVT and SDT, we investigated whether the science-learning motivation factor structure was best represented by four factors, including self-concept, intrinsic value, utility value, and sense of relatedness (Figure 1, Model 1) or by three factors, in which intrinsic value and utility value was combined as a general task value factor (Figure 1, Model 2). The theoretical foundation for the four-factor structure (Model 1) is that separateness of task values exists, where a student completes learning tasks either because of the enjoyment of learning (i.e., intrinsic value) or the usefulness of the learning (i.e., utility value). The three-factor model was based on literature in which researchers caution against unnecessary overlap among constructs when applying an integrative approach, thus striking a balance between capturing students' learning motivation as comprehensively but also as parsimoniously as possible (Linnenbrink-Garcia and Wormington, 2019) [15]. It is possible that facets of task intrinsic and utility values could overlap; for example, no matter which value underlies students' motivation (i.e., the enjoyment and/or the usefulness of the learning task), there may be a broad value that the student holds to learn the subject that is better represented by a general value (versus separate intrinsic and utility value) factors.

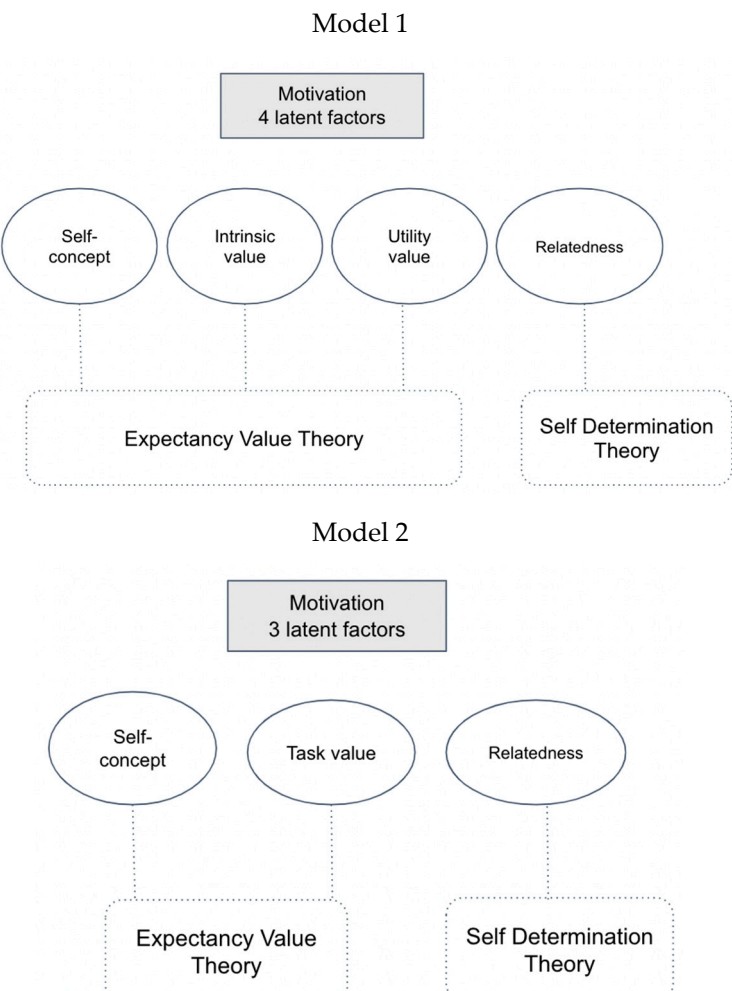

**Figure 1.** Two competing models of motivational constructs in TIMSS student science-learning motivation.

We did not examine a two-factor solution, one with SEVT and the other with SDT, because, based on the other TIMSS motivation studies, self-concept and task values were two factors that were widely drawn from SEVT together. Thus, from a theoretical perspective and empirical TIMSS motivation research, we did not examine a two-factor structure in this study.

*1.5. The Present Study*

This study aimed to investigate the factor structure of students' motivation in science learning by using theoretically aligned items from the TIMSS international database. There are studies examining teachers' perceptions of teaching (e.g., Cascarosa et al., 2021; and Wardat et al., 2022) [40,41], while others focus on students' perceptions of the learning process (e.g., Oo et al., 2023) [42], and the current study used TIMSS student questionnaires to examine students' perceptions of science-learning motivation specifically. The measurement model of student science motivation in the study integrated both SEVT and SDT frameworks to include a comprehensive set of motivation constructs. We systematically examined a baseline, three-factor, and four-factor model of science motivation to determine the best-fitting model using confirmatory factor analysis (CFA).

The following research question guided our study: Does a four-factor motivation structure (self-concept, intrinsic value, utility value, and relatedness) fit the TIMSS data better than a three-factor structure in which the task values are combined (self-concept, task value, and relatedness) and a one-factor structure (baseline model) for U.S. grade 8 students' motivation in science learning?

## 2. Materials and Methods

### 2.1. Dataset and Sample

Grade 8 U.S. student data from TIMSS 2015 datasets were used in this study. The International Association for the Evaluation of Educational Achievement (IEA) has administered TIMSS aiming to understand the impact of educational policies and practices across different education systems since 1995 (Mullis and Martin, 2015) [43]. TIMSS employed a nested two-stage randomized cluster sampling design to account for selection bias (LaRoche et al., 2016; and Martin et al., 2016b) [44,45]. In this study, we specifically focused on the data collected in the United States. We took sampling weights into consideration to ensure an accurate representation of the population. Sampling weights are the number of individuals in the population each respondent in the sample is representing. After excluding 66 completely missing cases, we retained a final sample size of 14,291 grade 8 students representing (with sampling weights) 3,656,774 students in the U.S., with a mean age of 14.27 years (SD = 1.79). Half (50%) of the sample (*N* = 7147) are boys. Among the 246 sampled schools, the number of sampled students per school ranged from 3 to 244. Around 89% of students always or almost always speak English at home, and 1.2% of students never speak English at home at all.

### 2.2. Measures

Each student participating in TIMSS completed a Student Questionnaire that asked about aspects of their home and school lives, including basic demographic information, characteristics of their home environment, school climate for learning, and self-perception and attitudes toward learning mathematics and science. Questionnaire items were selected from grade 8 student data files that reflected motivation constructs in this study. Item selection was based on definitions of constructs from the SDT and SEVT motivational frameworks (see Zhang and Bae, 2020 for review) [13]. Selected items were used to specify four motivation factors: self-concept, intrinsic value, utility value, and relatedness. Intrinsic value and utility value were combined as task value in the three-factor structure. Items measuring motivation constructs reflect students' self-reported levels of agreement with a 4-point Likert response scale, from 1 (disagree a lot) to 4 (agree a lot), so that a higher scale represents a higher level of agreement with the item. We followed the suggestion by Taber (2018) [46] that Cronbach's alpha should reach 0.70 for an instrument to have an acceptable level of internal consistency in science education (Table 1).

**Table 1.** Descriptive statistics for motivational constructs and corresponding items in TIMSS data.

| Factors (Alpha) | # | Items | Factor Loadings | Mean (SD [1]) |
|---|---|---|---|---|
| Self-concept (0.85) | 1 | I usually do well in science. | 0.86 | 3.32 (0.79) |
| | 2 | I learn things quickly in science. | 0.86 | 3.04 (0.90) |
| | 3 | I am good at working out difficult science problems. | 0.87 | 2.79 (0.97) |
| | 4 | My teachers tell me I am good at science. | 0.79 | 2.75 (1.00) |
| Intrinsic value—task value (0.94) | 5 | I enjoy learning science. | 0.89 | 3.12 (0.94) |
| | 6 | I learn many interesting things in science. | 0.78 | 3.34 (0.84) |
| | 7 | I like science. | 0.87 | 3.13 (0.96) |
| | 8 | I look forward to learning science in school. | 0.88 | 2.97 (0.99) |
| | 9 | Science is one of my favorite subjects. | 0.90 | 2.87 (1.07) |

**Table 1.** *Cont.*

| Factors (Alpha) | # | Items | Factor Loadings | Mean (SD [1]) |
|---|---|---|---|---|
| Utility value—task value (0.91) | 10 | I think learning science will help me in my daily life. | 0.83 | 3.11 (0.92) |
| | 11 | I need to do well in science to get into the university of my choice. | 0.89 | 3.24 (0.91) |
| | 12 | I need to do well in science to get the job I want. | 0.87 | 3.00 (1.03) |
| | 13 | It is important to learn about science to get ahead in the world. | 0.91 | 3.13 (0.93) |
| | 14 | Learning science will give me more job opportunities when I am an adult. | 0.91 | 3.20 (0.93) |
| | 15 | It is important to do well in science. | 0.87 | 3.44 (0.80) |
| Relatedness (0.80) | 16 | I like being in school. | 0.69 | 2.86 (0.88) |
| | 17 | I feel safe when I am at school. | 0.71 | 3.18 (0.85) |
| | 18 | I feel like I belong to this school. | 0.70 | 3.07 (0.95) |
| | 19 | I'm proud to go to this school. | 0.78 | 3.09 (0.95) |

[1]: SD represents standard deviation.

Self-concept: Based on the theoretical framework of SEVT and the comparison of self-concept versus self-efficacy in the first section, four items (items 1–4) related to competence beliefs were selected for self-concept, for example, "I usually do well in science" and "I learn things quickly in science". Cronbach's alpha reliability coefficient was 0.85 for self-concept ratings.

Task value—intrinsic value: As one of the task values in SEVT, intrinsic value consisted of five items (items 5–9) related to students' intrinsic enjoyment from learning science, for example, "I enjoy learning science" and "I look forward to learning science in school". Items that were related to interest, but not clearly aligned to the definition of this construct, such as "Science teaches me how things in the world work", were excluded, as this statement emphasizes the effect of science on the students more than students' inner interest in science. Cronbach's alpha reliability coefficient was 0.94 for intrinsic value ratings.

Task value—utility value: Six items (items 10–15) reflecting students' self-perceived value of learning science were included to represent utility value, for example, "It is important to learn about science to get ahead in the world" and "Learning science will give me more job opportunities when I am an adult". Cronbach's alpha reliability coefficient was 0.91 for utility value ratings.

Relatedness: Drawing from SDT, students' sense of relatedness consisted of four items (items 16–19) regarding a sense of belongingness, for example, "I like being in school" and "I feel like I belong to this school". Cronbach's alpha reliability coefficient was 0.80 for relatedness ratings.

### 2.3. Analyses

Confirmatory factor analysis (CFA) was employed using Stata Version 14.2 to examine the factor structure of motivation in the TIMSS dataset. CFA is often used in psychometric evaluations of instruments to determine the underlying structure of a scale and provide supportive evidence for the construct validity of a measure. This technique can both illuminate the number of underlying latent dimensions present in a set of items, as well as estimate the set of item-to-factor relationships, or loadings (Brown, 2006) [47]. When conducting a CFA, the number of factors that exist for a set of items and the extent to which each item is related to its designated factor are all specified a priori based on theory. Thus, in this study, the CFA tests how well the theoretical specification of motivation (three-factor vs. four-factor) fits with the TIMSS student data.

First, we examined a baseline model where all items are loaded on a single (one-factor) motivation factor (Model 0). This was treated as the baseline model, and served as the comparison point for the theoretically justified models tested next. We then compared two alternative multidimensional models of motivation: the four-factor model (Figure 1, Model 1) and the three-factor model (Figure 1, Model 2). The four-factor model tested whether or not the science-learning motivation model in TIMSS is best represented by four latent variables, including self-concept, intrinsic value, utility value, and relatedness. The three-factor model (Model 2) tested whether or not the science-learning motivation model in TIMSS is best represented by three latent variables, namely, self-concept, task value, and relatedness, in which intrinsic value and utility value are encompassed by general task value factor.

The maximum likelihood robust estimation was used to account for the non-independence of missing data by using method (mlmv) because of the small amounts of missingness automatically ($N = 66$). The CFA models were assessed using a set of absolute (fit from the obtained and implied covariance matrix), relative (fit from model test to a null model that specifies no latent variables), and comparative (relative fit of tested model compared with baseline model) goodness of fit (GOF) indices to determine the best-fitting model. The Root Mean Square Error of Approximation (RMSEA) adjusts for model complexity, making it sensitive to the number of parameters in the model (Brown, 2006) [47], with values of less than 0.06 considered a good fit (Hooper et al., 2008) [48]. The Comparative Fit Index (CFI: Bentler, 1990) [49] and the Tucker–Lewis Index (TLI: Tucker and Lewis, 1973) [50] compare the user-specified model to a baseline model, with values greater than 0.95 indicating good fit (Hu and Bentler, 1998, 1999) [51,52]. The Standardised Root Mean Square Residual (SRMR) is the square root of the difference between the residuals of the sample covariance matrix and the hypothesized covariance model, with values of less than 0.06 indicating good fit (Hu and Bentler, 1998, 1999; Kline, 2005) [51–53]. A probability value of $\alpha = 0.05$ for the chi-square ($\chi2$) test statistic is also reported but not used as a reference because chi-squared tests of model fit are often overly sensitive in large sample sizes, resulting in falsely rejecting an appropriate model (Gatignon, 2010) [54].

## 3. Results

### 3.1. Descriptive Statistics

Table 1 shows the descriptive statistics (mean and standard deviation) and factor loadings for all items. Factor loadings for self-concept ranged from 0.79 to 0.87, intrinsic value from 0.78 to 0.90, utility value from 0.83 to 0.91, and relatedness from 0.69 to 0.78. All the standardized factor loadings of the items were above the threshold limit of 0.6 (Hair et al., 1998, 2010) [55,56].

### 3.2. CFA Results for Motivation Structure Models

Below, the results of three competing measurement models of motivation are presented, including the baseline (Model 0), four-factor (Model 1), and three-factor (Model 2) models. Table 2 shows the results of the model GOF indices for all three models.

**Table 2.** Results of CFA comparing three competing models of motivation in TIMSS data.

| Model | # of Factors | $\chi2$ | df | RMSEA | CFI | TLI | SRMR | Range of Stdyx. Factor Loadings |
|---|---|---|---|---|---|---|---|---|
| Model 0: Baseline | 1 | 54,276 | 152 | 0.17 | 0.67 | 0.63 | 0.12 | 0.19–0.88 |
| Model 1: SC, IV, UV, R [1] | 4 | 5045 | 146 | 0.05 | 0.97 | 0.97 | 0.04 | 0.61–0.92 |
| Model 2: SC, TV, R | 3 | 32,428 | 149 | 0.13 | 0.80 | 0.77 | 0.09 | 0.56–0.89 |

[1]: SC = self-concept; IV = intrinsic value; UV = utility value; R = relatedness; TV = task value that includes intrinsic value and utility value.

Baseline (Model 0): The one-factor (baseline) model where all 19 items were loaded onto a single latent construct of motivation (Figure 2, Model 0) showed a poor model fit

to the data (RMSEA = 0.17, CFI = 0.67, TLI = 0.63, and SRMR = 0.12). An examination of the factor loadings showed that the intrinsic value items loaded well on the one-factor motivation latent construct (e.g., 0.85 for item 5 and 0.88 for item 7), whereas the sense of relatedness items showed low factor loadings (e.g., 0.19 for item 17).

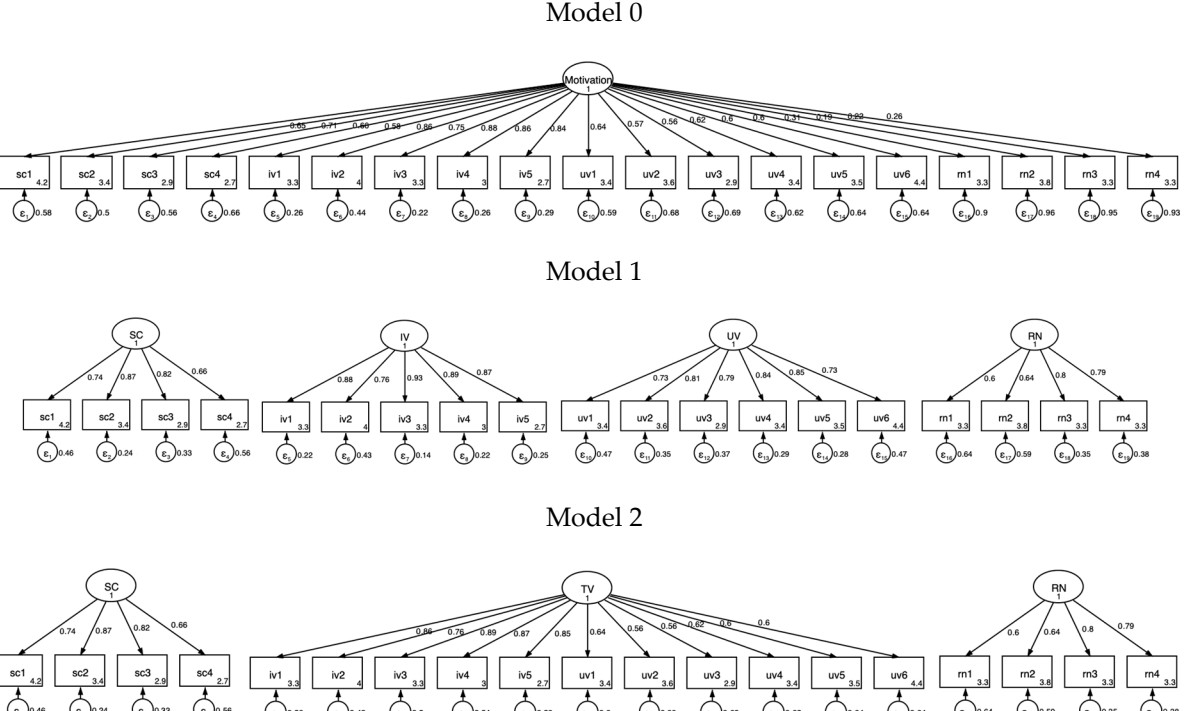

**Figure 2.** Motivation measurement models for baseline model, four-factor model, and three-factor model.

Four-factor (Model 1): The four-factor model included four latent variables representing self-concept (items 1–4), intrinsic value (items 5–9), utility value (items 10–15), and relatedness (items 16–19). The results showed a good fit of the four-factor model to the data (RMSEA = 0.05, CFI = 0.97, TLI = 0.97, and SRMR = 0.04). In addition to the substantial improvements over the baseline model in terms of the GOF indices, an examination of the factor loadings showed that all items loaded highly on their corresponding latent variables (ranging from 0.61 to 0.92).

Three-factor (Model 2): The three-factor model with intrinsic value and utility value combined together as a general task value factor also showed an improved model fit compared to the baseline model. However, the GOF indices for Model 2 did not provide evidence for an adequate model fit to the data (RMSEA = 0.13, CFI = 0.80, TLI = 0.77, and SRMR = 0.09). Thus, the results do not support a three-factor model where the intrinsic value and utility value are encompassed by a general task value factor.

## 4. Discussion

Drawing from SEVT and SDT, this study provides evidence for a four-factor model as an empirically supported and theoretically sound structure of science-learning motivation among middle school students (grade 8) using a large national dataset from TIMSS. In terms of the theoretical alignment of items and corresponding constructs, the findings suggest that the best-fitting science-learning motivation structure includes four unique dimensions, including self-concept, intrinsic value, utility value, and sense of relatedness. This examination of the motivation factor structure underlying the data from items available in the TIMSS student questionnaire extends our understanding of the dimensions of students' drive to learn science during an important developmental period in which

students are formalizing their attitudes towards academic activities and choices related to future professional careers.

First, we found that when intrinsic value and utility value are combined to represent a general task value factor (Model 2), the model showed a poor model fit to the data. This provides evidence that intrinsic value and utility value should be separated when examining students' science-learning motivation. Theoretically speaking, the separation of the task value components in the four-factor structural model makes students' motivation in science learning more specific, and further contributes to the CFA results. From the perspective of SEVT and SDT, learning motivation comes from internal drives and external stimuli; therefore, the four-factor structural model more accurately captures students' intrinsic and extrinsic aspects of science learning compared to the three-factor structural model. The results also echoed the need for theory integration (Rosli et al., 2022) [57]. Prior TIMSS motivation studies have also separated the intrinsic value and utility value when examining the relationship between students' motivation and science achievement (e.g., Lay and Chandrasegaran, 2016; Liou et al., 2020; and Wang and Liou, 2017) [11,17,58] and found that the strength of the two task values were different in predicting the outcomes. For example, Wang and Liou (2017) [11] found that intrinsic value (b = 5.23, *p* < 0.01) and utility value (b = 3.2, *p* < 0.05) were separately and positively associated with science achievement at the student level. However, some other studies that examined science-learning motivation with TIMSS data have used a general value in science as one of the indicators for science motivation. For instance, Leong and colleagues (2018) [18] constructed a general value in science with items from both the intrinsic value (e.g., I enjoy learning science) and utility value (e.g., I think learning science will help me in my daily life). Similarly, Yetisir (2014) [59] examined motivational factors that influence students' science achievement and used 20 items together, consisting of students' attitudes towards science. Based on the findings from our study, we recommend constructing science-learning motivation into four dimensions that model the intrinsic value and utility value as distinct factors.

The second major finding of this study concerns the clear identification of the sense of relatedness factor as a specific dimension of science-learning motivation among middle school students. This finding should be highlighted because few previous TIMSS studies of science-learning motivation have included students' sense of relatedness, as conceptualized in SDT. That is, although prior studies have cited SDT as a guiding framework, they used SDT to construct intrinsic and extrinsic motivation, as well as belief-related factors (e.g., Lay and Chandrasegaran, 2016; and Leong et al., 2018) [17,18]. Based on a systematic literature review, we propose that items available in the TIMSS student questionnaire align better with one of the basic needs underlying intrinsic motivation, namely, relatedness. The results from this study provide empirical evidence for the inclusion of this interpersonal component of student motivation in middle school, a period when peer relationships are shown to be more important in school learning experiences in diverse student populations (Brown and Larson, 2009; and Longobardi et al., 2016) [60,61]. Aligned with SDT, sense of relatedness in learning is the influence of teachers or peers in schools. Relatedness is shown to be an important indicator of social engagement in learning among secondary school learners. Importantly, the inclusion of relatedness is also in agreement with the recent science education reforms in the U.S., which put weight on students' social learning activities such as engagement in science argumentation with peers in class.

Taken together, our integrative approach aligns with contemporary work in which scholars argue that "students enter classrooms with myriad reasons to learn or disengage, and these reasons are not necessarily captured by one particular motivational theory" (Linnenbrink-Garcia and Wormington, 2019, p. 740) [15]. Our identification of motivation structures that are multifaceted opens the way for a more nuanced investigation of the patterns of science-learning motivation, using constructs that are empirically validated and theoretically aligned to related frameworks.

### 4.1. Practical Implications

While this study is largely aimed at advancing the understanding of the factor structure of science-learning motivation among middle school students from available TIMSS data, the results also have implications for practice. First, from an assessment perspective, this study supports the reliance on subtests designed to assess specific dimensions of task values (e.g., intrinsic value and utility value) for middle school students, as those that do not appear to reflect specific, distinguishable task values for students showed a poor model fit to the three-factor structure of science-learning motivation. Here, we are echoing a point made by task value researchers (e.g., Gaspard et al., 2017; Guo et al., 2016; and Kiuru et al., 2020) [62–64] that, for students in middle school, their learning motivation reflects a multifaceted underlying trait, and it seems best to rely on various assessments of science-learning motivation to explore individual differences and, potentially, to identify students with different dimensions of motivation, warranting different attention. The identification of unique motivation structures supports future research aiming to tease apart which and how different motivations underlie students' learning and achievement in science.

Next, with respect to instructional practices, this study supports the notion that students' sense of relatedness reflects a distinguishable type of motivation. There is increasing awareness of the importance of relatedness or belonging to learning motivation, and the crucial role of educators in facilitating students' sense of relatedness in school through positive teacher-to-student relationships and prosocial classroom climates. Previous research has provided recommendations for how this may be addressed. For instance, practical implications for educators to create this sense of relatedness include creating ongoing learning opportunities for students to engage in collaborative group work with clear purposes, roles, and shared goals (Anderman, 2003) [65]; teaching students strategies to strengthen interpersonal connections (e.g., providing constructive feedback to peers and working together towards mastery); and modeling caring and emotionally supportive behaviors for students to adopt towards an inquiry-based student-centered classroom climate (Moote, 2020) [66]. Our findings provide additional support for the idea that this interpersonal dimension is an important aspect of students' science-learning motivation.

### 4.2. Limitations

This study has several limitations that warrant consideration. We constructed science-learning motivation in this study in a general science domain; therefore, for future researchers conducting studies in specific science subjects, such as physics, chemistry, and biology, the items should be verified. Next, although our results regarding the factor structure of science-learning motivation can be considered more robust than previous TIMSS studies examining science motivation measurement models, in terms of the latent approach utilized in CFA analyses and the integration of multiple motivational theories, we acknowledge that these findings may lack generalizability given that students in this study were specific to the U.S. International large-scale datasets are always used to compare teaching and learning outcomes among various educational systems (e.g., Messina, 2023; Saal and Graham, 2023; and Zhang et al., 2023) [67–69], and the current study used U.S. data specifically; therefore, cultural differences should be taken into account in further studies. Considering the discrepancy of cultural and educational backgrounds and the complex nature of student motivation as it relates to their learning ecologies, future research, particularly studies that examine students' science-learning motivation from Eastern cultures, is warranted to generalize the findings in this study. Another limitation concerns the cross-sectional nature of TIMSS data which has limitations in tracking students' progress. Further tracking design studies are suggested. While we may infer that discrepancies existing in the factor structure of science-learning motivation could be borne out with a longitudinal assessment, this requires additional empirical investigation.

## 5. Conclusions

This study looks intensively at the dimensionality of science-learning motivation for middle school students, corresponding to the time period in which students typically formalize their attitudes towards academic activities and choices related to professional careers. Our results help us to understand more explicitly the nature of students' science-learning motivation during these formative years of schooling and science skill development. The results from the CFA analyses indicate that a four-factor structure representation of science-learning motivation emerged, comprising self-concept, intrinsic value, utility value, and sense of relatedness.

The issue of how motivation constructs are formed from TIMSS items requires closer examination and further explanation from both a statistical and theoretical standpoint based on an initial systematic literature review work (Zhang and Bae, 2020) [13], and, in view of the significance of integrating different motivational frameworks (Wigfield and Eccles, 2020) [70], a multifaceted motivation is required. This measurement study addresses those needs. The findings of this study contribute to the theoretical development of motivation specific to TIMSS data, and support the consistent, valid, and theoretically aligned use of motivation items from the TIMSS student questionnaire, that, together, can produce knowledge with which to inform educators and policy-makers in how to support students' motivation for success in science.

**Author Contributions:** Conceptualization, F.Z., C.L.B. and A.C.K.; methodology, F.Z., C.L.B. and M.D.B.; validation, F.Z. and M.D.B.; formal analysis, F.Z. and M.D.B.; writing—original draft preparation, F.Z.; writing—review and editing, C.L.B. and A.C.K.; supervision, C.L.B. All authors have read and agreed to the published version of the manuscript.

**Funding:** This research was funded by the Family–School Collaboration Professional Committee of China Educational Development Strategy Society, grant number JXXTKD2302, and the Open University of China, grant number Q21A0015.

**Institutional Review Board Statement:** Not applicable.

**Informed Consent Statement:** Not applicable.

**Data Availability Statement:** The datasets' information can be downloaded at: https://timssandpirls.bc.edu/timss2015/ (accessed on 15 May 2021).

**Conflicts of Interest:** The authors declare no conflict of interest.

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
