# Peer review of "Factor Structure of Student Science-Learning Motivation: Evidence from TIMSS U.S. Data"

_sustainability, doi:10.3390/su151713230_

Round 1

Reviewer 1 Report

The work presents the design of a questionnaire to assess the motivation of students towards science.

The questions posed in the questionnaire make it possible to assess the perception more than the students' own motivation, which should be assessed in other ways, such as checking if they are able of successfully carrying out tasks posed by the teacher (see and cite Cascarosa, E., Pozuelo, J. & Feringán, B. (2021). Old instruments in the physics and chemistry cabinet at Goya Secondary School. Analysis of their didactic use in teaching physics today. Culture and Education, 33 (3), 556-572, DOI: 10.1080/11356405.2021.1949113.).

On the other hand, the validation of the questions should be carried out by experts in the subject to verify that what is asked is what the students will interpret correctly.

For the rest, the text complies with the standard of the journal, it contains the appropriate sections, it is logical in its order and explanation, the English is correct and it is well cited.

Author Response

1.The questions posed in the questionnaire make it possible to assess the perception more than the students' own motivation, which should be assessed in other ways, such as checking if they are able of successfully carrying out tasks posed by the teacher (see and cite Cascarosa, E., Pozuelo, J. & Feringán, B. (2021). Old instruments in the physics and chemistry cabinet at Goya Secondary School. Analysis of their didactic use in teaching physics today. Culture and Education, 33 (3), 556-572, DOI: 10.1080/11356405.2021.1949113.).

Response: Thank you very much for the comments and the suggested paper. We read and agreed that there is a need to clarify the questionnaire questions and ability measures in the used datasets. The motivational instruments in the questionnaires measure the perception of the students in their learning motivation in science domain and there are also achievement tests that measure the students’ ability of successfully carrying out science tasks. This confirmatory factor analysis paper mainly tested the measures of students’ subjective motivational instruments based on the reviewed theories and the gaps in the TIMSS literature, and for the actual science ability of students, studies examining motivation and students’ science achievement should take the teachers’ rating into account. We have added this explanation in the paper: “There are studies examining teachers’ perceptions of teaching (e.g., Cascarosa et al., 2021; Wardat et al., 2022) while others focus on students’ perceptions of learning process (e.g., Oo et al., 2023), and the current study used TIMSS student questionnaires to examine students’ perceptions of science learning motivation specifically”, and cited the suggested reference accordingly. Thanks for helping make this clear to readers who are not familiar with TIMSS datasets.

  1. On the other hand, the validation of the questions should be carried out by experts in the subject to verify that what is asked is what the students will interpret correctly.

Response: Thanks for the concern about the validation of the questions carried out by experts in the subject. TIMSS datasets’ questionnaires have been validated by experts in the corresponding field from the official website and open books (see more here). But we do agree that this instruments used in this paper focus on general science domain and if future researchers conduct studies in specific science subjects, such as physics, chemistry, and biology, items should be verified and validated in the specific science domain. We added this as a caution and limitation at the end of the study (page 15, line 484).

  1. For the rest, the text complies with the standard of the journal, it contains the appropriate sections, it is logical in its order and explanation, the English is correct and it is well cited.

Response: Thank you for the encouraging comments and constructive suggestions from science education perspectives. We do appreciate that!

  1. Parts that must be improved:

Is the article adequately referenced? Are all the cited references relevant to the research?

Response: We have edited the references accourding to the guidelines and added relevant ones. Thank you!

Reviewer 2 Report

OverallThis study is a combination of qualitative and quantitative research. Based on the motivational factor structure in situational value expectation theory and self-determination theory, students were extracted from the TIMSS database, and student questionnaires were used to investigate students' motivation for scientific learning. The student questionnaire contains the motivation dimension in the above theory, comparing which of the three-dimensional and thinking structure of scientific learning motivation is more suitable for students' scientific learning motivation. The results show that the four-factor structure model can reveal the essential motivation of students for scientific learning. Based on the perspective of comparative research, this paper determines the structure of students' motivation for scientific learning, which is more rigorous and convincing than previous research.

Deficiency and recommendations

1.The data sample has a certain lag, because the cross-sectional data is used, so it is difficult to evaluate the basic situation of students' scientific learning motivation in the subsequent period, and further tracking design is required.

2.The reason why the analysis results of the three-factor structural model are not as good as the four-factor structural model should be explained in the discussion section. The author should explain the internal reasons from the perspective of SEVT and SDT.

3.As a highlight in the results of this study, student sense of relatedness, should be interpreted and analyzed in combination with the perspective of research theory(SEVT and SDT) to reflect its importance.

Please ask professionals to further refine the language of the manuscript.

Author Response

【Overall】This study is a combination of qualitative and quantitative research. 

Based on the motivational factor structure in situational value expectation theory and self-determination theory, students were extracted from the TIMSS database, and student questionnaires were used to investigate students' motivation for scientific learning. 

The student questionnaire contains the motivation dimension in the above theory, comparing which of the three-dimensional and thinking structure of scientific learning motivation is more suitable for students' scientific learning motivation. 

The results show that the four factor structure model can reveal the essential motivation of students for scientific learning. Based on the perspective of comparative research, this paper determines the structure of students' motivation for scientific learning, which is more rigorous and convincing than previous research.

【Deficiency and recommendations】

  1. The data sample has a certain lag, because the cross-sectional data is used, so it is difficult to evaluate the basic situation of students' scientific learning motivation in the subsequent period, and further tracking design is required.

Response: Thank you for pointing this out. We admitted that cross-sectional data has limitations in tracking students’ progress in nature, and we have added this as a caution in the paper. “The data used in the study is cross-sectional in nature and has limitations in tracking students’ progress. Further tracking design studies are suggested.”

  1. The reason why the analysis results of the three-factor structural model are not as good as the four-factor structural model should be explained in the discussion section. The author should explain the internal reasons from the perspective of SEVT and SDT.

Response: This is really a good point. Further explanations were added in the discussion: “Theoretically speaking, seperation of task value components in the four-factor structural model makes students’ motivation in science learning more specific, and further contributes to the CFA results. From the perspective of SEVT and SDT, learning motivation comes from internal drives and external stimulates, therefore, the four-factor structural model captures more students’ intrinsic and extrinsic aspects of science learning than the three-factor structural model. The results also echoed the need of theory integration (Rosli et al., 2022).” We appreciate this suggestion making the discussion connected to the frameworks.

  1. As a highlight in the results of this study, student sense of relatedness, should be interpreted and analyzed in combination with the perspective of research theory(SEVT and SDT) to reflect its importance.

Response: Similar to the comments above, we agree that interpretations regarding motivational frameworks should be combined in the discussion, and we modified as follows: “Aligned with SDT, sense of relatedness in learning is the influence of teachers or peers in schools. Relatedness is shown to be an important indicator of social engagement in learning among secondary school learners.” Thank you again for the constructive comments and suggestions!

  1. Comments on the Quality of English Language: Please ask professionals to further refine the language of the manuscript.

Response: We have asked professionals to further edit the whole paper accordingly and revised the inappropriate words and reference formats. Thank you!

Reviewer 3 Report

This paper addressed the need to contribute a more nuanced understanding of the motivation constructs of TIMSS datasets. I found the paper easy to read and the method clearly outlined. The results were logically analysed and the conclusions of interest to anyone who uses or interprets TIMSS datasets. 

The obvious limitation of the study was the total focus on US students. I would have liked to learn a little more about why this decision was taken, particularly as much of the interest in the TIMSS datasets is for it's value for comparison between nations.

I hope there are plans to duplicate the study with other country's data to see how generalisable these findings actually are.

This is an interesting paper and one that should prompt a good deal of interest.

Author Response

This paper addressed the need to contribute a more nuanced understanding of the motivation constructs of TIMSS datasets. I found the paper easy to read and the method clearly outlined. The results were logically analysed and the conclusions of interest to anyone who uses or interprets TIMSS datasets. 

Response: Thank you very much for the review and encouraging comments!

The obvious limitation of the study was the total focus on US students. I would have liked to learn a little more about why this decision was taken, particularly as much of the interest in the TIMSS datasets is for it's value for comparison between nations. I hope there are plans to duplicate the study with other country's data to see how generalisable these findings actually are.

Response: This is really an important point. We totally agree and appreciate this suggestion. We added the following as a limitation: “International large-scale datasets are always used to compare teaching and learning outcomes among various educational systems (e.g., Messina, 2023; Saal & Graham, 2023; Zhang et al., 2023), and the current study used U.S. data specifically, therefore, cultural differences should be taken into account in further studies.” We plan to duplicate the study with data from an Eastern country in the future and also modified the data information in the title so that it could give readers a clearer guidance. Thank you!

This is an interesting paper and one that should prompt a good deal of interest.

Response: Thanks again for the constructive comments! We appreciate that.

Round 2

Reviewer 1 Report

The authors have completed the changes proposed. Now it is publicable.